# Post-Translational Modifications of Circulating Alpha-1-Antitrypsin Protein

**DOI:** 10.3390/ijms21239187

**Published:** 2020-12-02

**Authors:** Urszula Lechowicz, Stefan Rudzinski, Aleksandra Jezela-Stanek, Sabina Janciauskiene, Joanna Chorostowska-Wynimko

**Affiliations:** 1Department of Genetics and Clinical Immunology, National Institute of Tuberculosis and Lung Diseases, 01-138 Warsaw, Poland; ulka100@gmail.com (U.L.); sf.rudzinski@gmail.com (S.R.); jezela@gmail.com (A.J.-S.); sabinajanciauskiene@gmail.com (S.J.); 2Member of the German Center for Lung Research DZL, Department of Respiratory Medicine, Biomedical Research in Endstage and Obstructive Lung Disease Hannover BREATH, 30625 Hannover, Germany

**Keywords:** alpha-1-antitrypsin, AAT, *SERPINA1*, chronic obstructive pulmonary disease, protease inhibitor, S-nitrosylation, glycosylation, oxidation, carbamylation, homocysteinylation

## Abstract

Alpha-1-antitrypsin (AAT), an acute-phase protein encoded by the *SERPINA1* gene, is a member of the serine protease inhibitor (SERPIN) superfamily. Its primary function is to protect tissues from enzymes released during inflammation, such as neutrophil elastase and proteinase 3. In addition to its antiprotease activity, AAT interacts with numerous other substances and has various functions, mainly arising from the conformational flexibility of normal variants of AAT. Therefore, AAT has diverse biological functions and plays a role in various pathophysiological processes. This review discusses major molecular forms of AAT, including complex, cleaved, glycosylated, oxidized, and S-nitrosylated forms, in terms of their origin and function.

## 1. Introduction

The amino acid sequence determines the three-dimensional structure of each protein, although each protein’s function is largely modulated by post-translational modifications (PTMs). The term PTM indicates changes in the polypeptide chain because of the addition or removal of distinct chemical moieties to amino acid residues, proteolytic processing of the protein, or interactions between the protein and other substances [1,2]. Protein PTMs are involved in most cellular processes including metabolic regulation and defense against pathological insults. Therefore, recognition and analyses of protein PTMs, as well as efforts to understand their biological significance, are fundamental considerations in experimental and clinical science. Because PTMs are implicated in the development of various diseases, there are increasing efforts to identify connections between altered PTMs and the development of specific pathologies.

Alpha-1-antitrypsin (AAT) is the prototypical member of the serine protease inhibitor (SERPIN) superfamily (Figure 1) [3]. The AAT coding gene, *SERPINA1*, is 12.2 kb in length and composed of seven exons (Ia, Ib, Ic, and II–V) and six introns. It resides in a gene cluster that includes α1-antichymotrypsin, AAT pseudogene, cortisol-binding globulin, and protein C inhibitor [4]. *SERPINA1* is extremely polymorphic, such that more than 150 single-nucleotide polymorphisms have been reported. The *SERPINA1* pathogenic variants are known to cause AAT deficiency (AATD), an autosomal-codominant disorder. There are several clinically relevant genetic variants of the AAT protein with low expression levels, which polymerize spontaneously or show minimal/no inhibitory activity [5,6]. Normal variants of AAT also possess considerable conformational flexibility that permits the generation of various molecular forms (e.g., polymeric, cleaved, oxidized, and complexed with other substances) [7,8]. Accordingly, AAT demonstrates broad biological activity and participates in various pathophysiological processes.

This review discusses some of the most common types of PTMs currently studied in AAT research, focusing on their origin and functionality (Figure 2).

## 2. Glycosylation of AAT

Protein glycosylation encompasses a diverse array of sugar-moiety additions to proteins and is a major type of PTM with important effects on protein folding, conformation, distribution, stability, and activity. Carbohydrates in the form of asparagine (Asn)-linked (N-linked) or serine/threonine-linked (O-linked) oligosaccharides are major structural components of many secreted proteins [10,11]. Human AAT is a glycoprotein with a carbohydrate content of approximately 15% [12]. AAT demonstrates Asn-linked glycosylation at three specific sites on its polypeptide backbone. Two of these sites are present on Asn 46 and Asn 83, encoded within exon II, and one site is located on Asn 247, encoded within exon III [13]. Core glycans N-acetylglucosamine are directly attached to the protein via Asn residues. Subsequently added glycans include galactose, mannose, fucose, and sialic acid, which form bi-, tri-, and tetra-antennary branching structures. There are nine known glycoforms of AAT, M0–M8, determined by motifs attached to Asn83. Asn46 and Asn247 have di-antennary glycans (M6 and M8), tri-antennary glycans (M4 and M7), and tetra-antennary glycans (M2). M8 and M7 have five amino acid deletions in their sequences [14]. Although there are many different glycoforms of AAT, few studies have addressed AAT glycosylation.

Glycosylation increases protein stability by protecting against proteolysis and degradation [15,16]. Furthermore, glycans extend the half-life of secreted AAT protein and prevent its aggregation [17]. The glycosylation of AAT is important in its anti-protease and immunomodulatory functions. For example, aerosolized transgenic AAT, which differs from native AAT in terms of glycosylation, has lower ability to inhibit neutrophil elastase activity [18] compared to aerosolized native AAT [19].

Previous studies have shown that cytokines regulate the extent of AAT glycosylation. For example, interferon-γ, interleukin (IL)-1, IL-6, tumor necrosis factor (TNF)-α, and transforming growth factor-β all either enhance or reduce the numbers of branched oligosaccharides in AAT [20]. Furthermore, a study comparing AAT production by hepatoma and lung epithelial cells in response to oncostatin M (an analog of IL-6) demonstrated upregulation of AAT production in both cell types, although the glycosylation pattern differed [21]. These findings highlight the importance of AAT as an acute phase protein, the expression of which increases considerably during inflammation. It also modifies glycoforms in response to inflammation.

Changes in AAT glycan expression in the serum were observed in patients with community-acquired pneumonia, both during the acute phase of disease and at the time of resolution. Previous studies have demonstrated considerable impacts of N-glycans on the immunomodulatory functions of AAT, specifically through adjustments of interactions with IL-8 [22]. In liver biopsy tissue specimens, IL-6 expression was significantly positively correlated with the expression of the *FUT6* gene, which encodes the protein responsible for outer arm fucosylation. The resulting gene expression changes led to elevated serum levels of tri-antennary, tri-sialylated, and mono-fucosylated glycans of AAT-A3F in patients with non-alcoholic steatohepatitis, compared to patients who had non-alcoholic fatty liver. Serum AAT-A3F levels were significantly elevated in the context of pathological conditions such as hepatic fibrosis, intrahepatic inflammation, and hepatocyte ballooning. Notably, AAT-A3F has shown promise as a noninvasive marker of non-alcoholic steatohepatitis [23,24]. Another study showed that upregulation of core-fucosylated AAT could be a marker for hepatocellular cancer, while antennary fucosylation of AAT may be a marker for inflammation, particularly in patients infected with hepatitis B virus [14,25]. It is possible that the diverse charges in AAT glycopeptides associated with the sialic acid/galactose linkage of the glycan motif can differentiate between early stage hepatocellular carcinoma and cirrhosis [26]. AAT glycosylation patterns have been analyzed in patients with different cancer types (i.e., small-cell lung cancer and non-small-cell lung cancer) and subtypes (i.e., lung adenocarcinoma and squamous cell lung cancer) [27]. Those data revealed that the galactosylated AAT glycoform can be used to discriminate non-small-cell lung cancer from benign pulmonary nodules. By contrast, high serum AAT fucosylation was very specific for adenocarcinoma and strongly correlated with the stage of adenocarcinoma. Moreover, poly N-acetyllactosamine (poly LacNAc) was elevated in the sera of patients with small-cell lung cancer, compared to healthy controls [27]. Those findings imply that poly LacNAc is essential for tumor invasion and metastasis as a modulator of interactions between cancer cells, as well as between cancer cells and the extracellular matrix.

A recent study found highly sialylated M0 and M1 AAT glycoforms in the sera of patients with coronavirus disease 2019 (COVID-19), which might indicate a specific host response to virus-induced inflammation. In that study, the authors found a correlation between COVID-19 severity and the IL-6:AAT ratio. Patients with greater COVID-19 severity had a higher IL-6:AAT ratio, compared to patients who had community-acquired pneumonia [28]. AAT modulates activities that result in downstream IL-6 inhibition, which is implicated in COVID-19 pathogenicity [29]. Moreover, AAT has regulatory roles in the coagulation cascade [9] and could inhibit immunothromboses via elastase inhibition [30]. Notably, inflammatory dysregulation and coagulopathies have been reported in patients with varying levels of COVID-19 severity [31]. These observations support the exploration of anti-inflammatory roles for AAT by means of a clinical trial for the treatment of patients with COVID-19 using AAT supplementation therapy. This therapy is currently approved for patients with emphysema who have severe inherited AATD [28]. We presume that the levels and diversities of AAT glycosylation forms might be implicated in COVID-19 pathogenicity.

## 3. Protease Complexed and Cleaved Forms of AAT

Human AAT is a major protease inhibitor that forms a stable complex with a range of serine proteases. AAT is the most rapid inhibitor of human neutrophil elastase, proteinase 3, and cathepsin G. Its second-order constants of association with these proteases are 6.5 × 10^7^, 8.1 × 10^6^, and 4.1 × 10^5^ M^−1^ s^−1^, respectively [32,33]. Crystallographic studies have revealed that the binding of serine proteases to AAT cleaves the reactive center loop of AAT, resulting in complex formation whereby the protease is moved to the opposite end of the AAT molecule (Figure 3) [34,35].

This reaction irreversibly impedes activity for both the protease and AAT. New findings support the ability of AAT to inhibit a broad range of serine proteases, as well as other classes of proteases (e.g., metalloproteases and cysteine-aspartic proteases). Among those, cysteine-aspartic proteases are caspases, calpain-1, kallikreins 7 and 14, and TNF-α-converting enzyme ADAM-17 [34,36]. Importantly, a recent study identified AAT as a novel inhibitor of *TMPRSS2*, which plays a role in the cellular entry of several coronaviruses, including severe acute respiratory syndrome (SARS)-coronavirus (CoV)-2, SARS-CoV, and Middle East respiratory syndrome-CoV, as well as influenza viruses [37,38,39,40]. AAT impedes SARS-CoV2 cellular entry and prevents the main clinical complications of severe COVID-19, such as acute inflammation and acute respiratory failure [41]. Therefore, it could be hypothesized that the geographical distribution of pathogenic variants of AAT protein is to a certain extent linked to differences in COVID-19 epidemiology and severity throughout Europe [42]. Furthermore, a Food and Drug Administration-approved AAT augmentation therapy, used for patients with AATD and emphysema, has potential value in managing SARS-CoV-2 infection and an aberrant immune response. Two clinical trials (NCT04495101 and NCT04547140) are underway to evaluate the safety and efficacy of intravenous infusion of AAT in patients with COVID-19.

According to the MEROPS database (http://merops.sanger.ac.uk/index.shtml last accessed on 29.10.2020), AAT interacts with non-specific proteases without forming stable inhibitor complexes. In these instances, proteases recognize the reactive center loop of AAT between residues 352 and 365 as a substrate and hydrolyze this loop, but do not form stable AAT–protease complexes. Several non-serine proteases inactivate AAT via cleavage. These include cathepsin; stromelysins 1 and 3; matrix metalloproteases 1, 3, 7, 9, and 11; matrilysin-1; collagenase-1; and gelatinase B [43]. This inactivation of AAT by proteolysis may shift the protease–anti-protease balance in favor of proteolysis. Various viruses, including corona and influenza viruses, are sensitive to the respiratory protease/antiprotease balance. Therefore, strategies preventing nonspecific AAT inactivation and maintaining the protease–antiprotease balance may serve as targets to prevent viral infections [44].

## 4. AAT Peptides

Various studies have shown that proteolytic cleavage of native AAT produces larger N-terminal and shorter C-terminal fragments bound to the cleaved complex [45]. Notably, in contrast to biochemical and structural data, variants of AAT C-fragments have been identified in human tissues and body fluids [46]. We recently identified novel short transcripts of the *SERPINA1* gene, which are differentially expressed in tissues and cells. We have also provided experimental evidence that the expression of short transcripts can be regulated, and that C-terminal peptides of AAT can be secreted [47].

The physiological functions of these peptides have also been reported, including natural killer cell suppression (CRISPP peptide) [48], extracellular matrix protection (SPAAT peptide) [49], pro- or anti-inflammatory immune-modulating functions [50,51], and inhibition of human immunodeficiency virus entry (VIRIP peptide) [52]. We previously reported that the C-terminal fragment of AAT forms amyloid-like fibrils that activate human monocytes. Moreover, this fragment is present in atherosclerotic plaques [53] and the lungs of patients with chronic obstructive pulmonary disease (COPD) [54]. A putative connection has been suggested among matrix metalloprotease activity, generation of C-terminal peptides of AAT, and tumor progression [55]. Finally, the Y105 C-terminal peptide of AAT, CSIPPEVKFNKPFVYLI, has been characterized as a siRNA endoporter. Therefore, the Y105 peptide is a promising candidate for targeted therapy [56].

Various clinical studies have suggested that C-terminal peptides of AAT may be useful as biomarkers. For example, in patients with severe sepsis, serum levels of the 42-amino acid C-terminal peptide of AAT, CAAP48, were significantly elevated. Thus, this peptide may have value as a diagnostic biomarker for sepsis (Figure 4) [57,58].

Likewise, upregulation of AAT and its fragments has been observed in chronic kidney diseases and could serve as a biomarker [59,60]. A recent study suggested that the C-terminal peptide of AAT may be a predictive biomarker for acute exacerbations in patients with idiopathic pulmonary fibrosis [61]. Moreover, AAT fragments might serve as biomarkers in certain spinal cord pathologies, such as chronic neuropathic pain, which involves the somatosensory nervous system. Using consecutive enzymatic reactions and mass spectrometry, Bäckryd et al. analyzed PTMs of 18 isoforms of AAT in cerebrospinal fluid. They identified two N-truncated AAT peptides, one downregulated (AT5106, 19.4 kDa) and another upregulated (I_AT111, 38.9 kDa), which could be used to distinguish between patients with chronic neuropathic pain and healthy controls [62]. Moreover, statistical analyses revealed I_AT111 as an independent predictor of chronic pain intensity.

The protein fragments that result from protease-associated cleavage of AAT might serve as unique signatures for specific proteases. Degradomics–peptidomics profiling of biological fluids might reveal unique AAT peptides with diagnostic and/or prognostic value.

## 5. Polymers of AAT

AAT has a structure that is a hallmark of the SERPIN superfamily, comprising three β-sheets (A–C), nine α-helices (A–I), and a reactive center loop bait sequence that acts as a pseudo-substrate for the target protease (Figure 3 and Figure 5). Structural lability of the central β-sheet (A) is required for inhibitor function and is essential for polymer formation. The mechanisms by which AAT forms polymers are the focus of ongoing investigations. Several genetic variants of AAT, such as the Z (Glu342Lys) variant, have a propensity for the formation of ordered linear polymers. The classic pathway of SERPIN polymerization suggests that the Z mutation of AAT disrupts the relationship between the reactive center loop and β-sheet A [63]. An opening in the s4A cavity allows for the creation of a sequential β-strand linkage between the reactive center loop of one SERPIN and β-sheet A of another SERPIN, leading to the formation of a dimer and subsequent polymers [6,36,64,65]. Two additional models for AAT polymerization have been proposed, both of which suggest that AAT assembly pathways could arise from structurally and/or dynamically distinct polymerogenic intermediates. The structures of pathological polymers that accumulate in patients are unclear.

However, a recent study concerning structural characterization of AAT polymers obtained from post-transplant liver tissue of patients with the ZZ variant of AAT greatly favored the C-terminal disulfide (Cterm) domain swap as the structural basis for pathological Z-AAT polymers [64]. The C-terminal disulfide domain swap is presumably the basis of pathological polymers of Z-AAT, which are present in more than 95% of patients with severe AATD. However, the structural aspects of polymers resulting from other mutants of AAT, such as Siiyama, Mmalton, and King’s, have not been elucidated. A recent study revealed Trento (Glu75Val), a novel variant of AAT [66]. The Trento variant displayed electrophoretic profile polymers other than Z-AAT. Electrophoresis, immunological studies, and molecular modeling results demonstrated that these polymers were caused by the loss of native protein stability. The Trento variant might be responsible for the transition of AAT to latent and/or polymeric states in plasma, similar to other SERPINs (e.g., antithrombin) [67]. Native, wild-type AAT can form polymers while interacting with hydrophobic molecules, such as heme and fatty acids. However, the clinical significance of such polymerization remains unclear.

Approximately 2–5% of Europeans are heterozygous for the Z and wild-type M alleles. Remarkably, a recent study elegantly demonstrated that the Z and M forms of AAT can copolymerize [68]. These data showed that Z-AAT can form heteropolymers with polymerization-inert variants in vivo, which has implications for liver disease in heterozygous individuals.

The Z-AAT variant is the primary cause of severe AATD and liver disease due to polymer formation in the endoplasmic reticulum of hepatocytes and hepatocyte damage. Extracellular polymers of AAT are also present, which might be pro-inflammatory and stimulate neutrophil chemotaxis [69]. This pro-inflammatory function, together with the impaired anti-serine protease activity of Z-AAT, may enhance disease susceptibility and promote the development of emphysema [70]. Notably, AAT polymers may serve as templates for the binding of bacterial or yeast-related products, leading to the generation of pro-inflammatory and chemotactic forms of the polymer [71]. Moreover, oxidative stress can greatly accelerate the intracellular polymerization of Z-AAT. This process may occur in different cell types (e.g., pancreatic, endothelial, and monocytes) and may contribute to panniculitis and systemic vasculitis associated with Z-AAT [72]. Disease susceptibility may be directly related to the accumulation of aggregated AAT protein, although it may also depend on the rate of polymer clearance by autophagy and non-proteasomal degradation, which could influence the clinical phenotype [73].

The presence of AAT polymers in human alveolar macrophages, and their effects on lung inflammation, have recently been demonstrated [74] both in patients with COPD who have AAT-deficient PiZZ or PiZI variants and in smokers with PiMM who are otherwise healthy or have COPD.

The silencing of Z-AAT protein production might provide an effective therapeutic intervention towards prevention and/or treatment of AATD-induced liver disease [75,76]. Attempts have been made to prevent nascent polymer accumulation and to facilitate reduction of preexisting polymers with an RNAi trigger to selectively degrade Z-AAT mRNA [77,78]. Thus far, there remains no disease-specific therapy for hepatic manifestations of AATD. Liver transplantation remains the definitive treatment for advanced liver disease [79].

## 6. Complexed Forms of AAT

AAT interacts with various ligands (Figure 2). For example, AAT binds to secreted enteropathogenic *Escherichia coli* proteins EspB and EspD [80]; *Cryptosporidium parvum* [81], a protozoan parasite; free heme [72]; various lipid moieties (e.g., low-density lipoprotein (LDL), high-density lipoprotein (HDL), and fatty acids) [82,83,84], chemokines IL-8 and LTB4 [85,86], complement factors [87], and heat shock proteins [88] (Figure 2). Important, AAT interacts with pre-prohepcidin and prohepcidin, but not with the mature form of hepcidin, a controller of iron homeostasis [89]. This interaction may protect prohepcidin from cleavage by furin, a serine protease responsible for hepcidin maturation [90].

Some AAT complexes have been linked to specific clinical conditions. For example, AAT complexed with the kappa light chain of immunoglobulin was detected in the plasma of patients with myeloma and Bence–Jones proteinemia [90]. Furthermore, AAT/factor XIa and AAT/heat shock protein-70 complexes were found in patients with diabetes [88], while AAT/immunoglobulin A complexes were detected in the synovial fluid of patients with rheumatoid arthritis (RA), systemic lupus erythematosus, and ankylosing spondylitis [91]. Human kallikrein 3, a prostate-specific antigen, was complexed with AAT in serum samples from patients with high prostate-specific antigen concentrations [92]. Moreover, AAT associates with HDL, which likely explains the anti-elastase activity ascribed to HDL [93]. In a mouse model of elastase-induced pulmonary emphysema, AAT binding to HDL enhanced the protective effects of AAT [94]. Serum levels of AAT/oxidized LDL complexes were high in smokers and decreased after smoking cessation [95]. Kotani et al. showed that AAT/LDL complexes were positively associated with adiponectin and HDL cholesterol in women without metabolic syndrome, but not in women with metabolic syndrome [96]. To date, AAT complex forms with other substances have received little attention in clinical and experimental research.

## 7. Oxidized Forms of AAT

Oxidation of critical methionine residues in AAT, specifically at positions 351 and 358 (pivotal for establishment of the proteinase-inhibitor complex) [97], generates oxidized protein. This reaction is presumed to have pro-inflammatory consequences because inactivation of AAT inhibitory activity favors proteolysis, inflammatory processes, and connective tissue degradation. Oxidative modifications of AAT protein are induced by cigarette smoke components, as well as oxidants and enzymes (e.g., myeloperoxidase released by cells at sites of inflammation). The oxidized forms of AAT have minimal or no anti-elastase activity. They have been proposed to serve as oxidative stress markers [98] and potential biomarkers for the harmful effects of smoking. Moreover, these forms are important in COPD development [99]. Higher levels of serum-oxidized AAT were found in smokers with COPD than in nonsmokers with COPD or healthy controls [99]. Furthermore, partially inactivated oxidized AAT was detected in bronchoalveolar lavage fluid from smokers, but not from nonsmokers. The oxidation of AAT is considered an important promoter of proteinase inhibitor imbalance in the lungs [100]. Stephenson et al. reported that oxidized AAT forms a substantial fraction of total AAT in the lungs of patients with human immunodeficiency virus, suggesting that the oxidation of AAT may contribute to emphysema development in these patients [101]. Moreover, Izumi-Yoneda et al. suggested that AAT oxidation in the amnion is associated with premature rupture of the fetal membranes [102]. Ueda et al. reported the presence of oxidized AAT in serum samples from patients with inflammatory diseases [103]. Specifically, oxidized AAT has been detected in patients with rheumatic diseases, bronchiectasis [104,105], and heart failure [106]. Notably, Jamnongkan et al. identified oxidized AAT as a potential risk indicator for opisthorchiasis-associated cholangiocarcinoma [107].

Experimental studies have shown that oxidized AAT promotes the release of monocyte chemoattractant protein-1 and IL-8 from A549 pulmonary epithelial cells and normal bronchial epithelial cells. This process may contribute to COPD pathogenesis by causing a functional deficiency and generating a pro-inflammatory form of AAT [89]. Alam et al. demonstrated that cigarette smoke induces oxidation and subsequent polymerization of Z-AAT, thereby rendering AAT inactive as an anti-elastase and creating a pro-inflammatory environment [108]. These findings suggest that major risk factors for COPD (e.g., cigarette smoke and Z-AAT deficiency) have a synergistic effect, rather than independent or additive effects. It is important to note that, similar to the native form, oxidized AAT interacts with other molecules. Oxidized AAT complexed with IgA was detected in patients with ankylosing spondylitis [109] and RA [91]. Furthermore, oxidized AAT/LDL complexes were found in atherosclerotic lesions [110]. The pathophysiological roles of these interactions remain unknown.

Despite a loss of inhibitory activity, oxidized AAT inhibits lipopolysaccharide-induced production of TNF-α and IL-1β cytokines in vitro. It also inhibits cigarette smoke-induced inflammation in vivo. However, other studies have demonstrated that the oxidized AAT expresses anti-inflammatory effects. An important example of this involves the smoking mouse model. These animals demonstrate a TNF-α response with downstream inflammation, proteinase release, and tissue destruction, which lead to emphysematous changes. Importantly, administration of oxidized AAT blocks these effects [111]. However, an experimental in vivo mouse model involving site-directed mutagenesis (to replace AAT M351 with valine (V) and M358 with leucine (L)) revealed maintenance of antiprotease activity under oxidant stress despite abrogation of wild-type AAT function, indicating a potential novel strategy for treatment of AATD [112].

In agreement with these findings, native and oxidized forms of AAT are equally effective for preventing acute liver and lung injury in vivo [113,114]. Taken together, the current data suggest that oxidative modifications attenuate the serine protease-inhibiting effect of AAT, although they do not entirely abolish the anti-inflammatory effects. Hence, AAT lacking anti-protease activity can efficiently have anti-inflammatory functions. This implies that the anti-inflammatory and anti-protease functions of AAT can be fully independent.

## 8. S-Nitrosylated Form of AAT

Protein S-nitrosylation involves covalent attachment of a nitric oxide (NO) group to cysteine residues via direct interactions between reactive nitrogen species and protein thiol residues, or via nitrosylation by S-nitrosoglutathione [115]. The only cysteine residue of AAT (Cys232) undergoes S-nitrosylation (S-NO-AAT) in the inflammatory milieu [116]. The reverse process, protein denitrosylation, is driven by two enzyme systems: glutathione/S-nitrosoglutathione reductase and thioredoxin/thioredoxin reductase [117]. Denitrosylation can also be self-induced by the S-NO-protein, as observed in S-NO-AAT, via transfer of the NO molecule to cellular targets with free thiols [116,118].

Some data suggest that human plasma AAT is readily S-nitrosylated under physiological conditions, and that this nitrosylation is 10-fold more efficient than the reaction between albumin and glutathione. The bacteriostatic effects of S-NO-AAT enabled via transnitrosylation was 20- to 3000-fold more effective than for other S-nitrosylated albumins and glutathione [118]. Moreover, S-NO-AAT appears to stimulate a significantly stronger anti-bacterial response in macrophages, compared to native AAT or S-nitrosoglutathione [116]. In addition, S-NO-AAT can prevent hepatocyte apoptosis in rats [118,119]. The beneficial effects of S-NO-AAT are associated with its antioxidant activity and ability to induce heme oxygenase-1 expression [119].

A recent study reported that S-nitrosylation facilitates the antibacterial activity of AAT by promoting its ability to activate innate immune cells [116]. Although the precise proportion of S-NO-AAT generated in vivo is not known, nor the ratio between AAT/S-NO-AAT, the authors of this latter study postulated that during infection, AAT becomes S-nitrosylated, and thus can assist in the reduction of the bacterial burden. S-NO-AAT seems to cause resting macrophages to exhibit a pro-inflammatory and antibacterial phenotype, including release of inflammatory cytokines and induction of inducible nitric oxide synthase. A similar outcome was observed in other nitrosylated circulating proteins, including albumin [120], and hemoglobin [121].

At infection sites, immune cells express inducible nitric oxide synthase (iNOS), followed by production of nitric oxide (NO) [122] that can act as a signaling molecule by promoting S-nitrosylation (S-NO) of both host and pathogen proteins. Protein S-nitrosylation is involved in gene transcription and protein function, as well as in inflammatory and cell survival pathways [123]. S-NO-hAAT gains the ability to directly eliminate bacteria [118].

## 9. Carbamylated and Homocysteinylated AAT

Protein carbamylation is a PTM partially caused by exposure to the urea dissociation product cyanate, as well as inflammation, diet, smoking, and environmental factors [124]. The net result of the carbamylation reaction is the addition of a “carbamoyl” moiety (-CONH2) to a functional group. The 34 lysine residues in the AAT protein provide a suitable target for carbamylation [125]. Notably, carbamylated AAT has been identified in carbamylated fetal calf serum [126].

Patients with RA have been found to make autoantibodies toward the carbamylated form of AAT in the synovial fluid. These antibodies can be observed years before disease onset, and may predict the development of RA in patients with arthralgia. Consequently, carbamylated AAT is currently being developed as an antigenic biomarker for RA [127]. Carbamylated AAT may also serve as a marker of the loss of AAT protection [128]. The latter may provide a justification for exploring AAT therapy for RA patients.

Homocysteine thiolactone is a cyclic thioester of homocysteine that contributes to the toxicity of this amino acid. Homocysteine thiolactone spontaneously reacts with protein lysine residues, leading to altered target protein properties and immune response induction. In a recent study, Colasanti et al. used reverse-phase nanoliquid chromatography and tandem mass spectrometry to confirm that homocysteinylation is a PTM of AAT. In patients with RA, rheumatoid factor and anti-citrullinated protein antibodies are most frequently detected [128]. However, Colasanti et al. observed that a large number of patients with RA displayed anti-homocysteine-AAT autoantibodies, which might serve as biomarkers for RA.

## 10. Conclusions

The conformational polymorphism of AAT protein confirms the complexity of its biological functions. Importantly, AAT interacts with other molecules, which may lead to oxidation, degradation, complex formation, self-assembly, or additional modifications. Some of these changes result in acquired deficiency of native AAT protein; however, they may also generate new molecular forms with potent biological activity, which may have prognostic significance and predict therapeutic response. Unfortunately, no standardized methods are currently available to assess the levels and clinical roles of post-translationally modified forms of AAT.

## Figures and Tables

**Figure 1 ijms-21-09187-f001:**
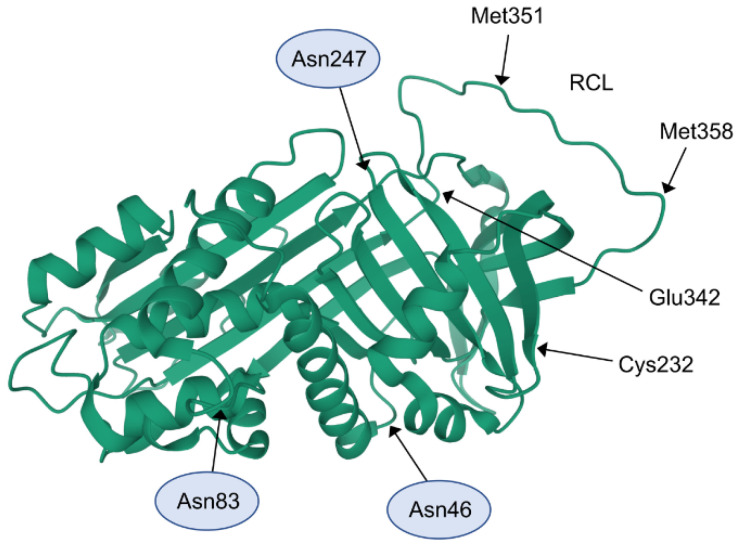
Schematic structure of native alpha-1-antitrypsin. The reactive center loop (RCL), three β-sheets, and nine α-helices are depicted. The amino acids discussed in the text are marked in the diagram as Asn46, Asn83, and Asn247 (surrounded by circles) are glycosylation sites; Met351 and Met358 are residues that undergo oxidation; Cys232 is an S-nitrosylation site; and Glu342 is the site of Glu342Lys substitution. The alpha-1-antitrypsin (AAT) native diagram was prepared based on data obtained from the SWISS-MODEL repository (https://swissmodel.expasy.org last accessed on 23 November 2020).

**Figure 2 ijms-21-09187-f002:**
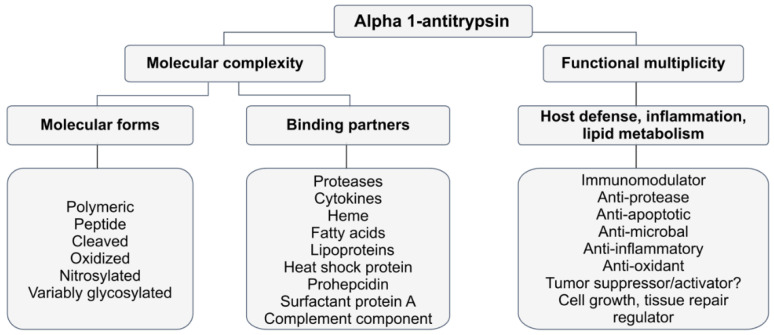
The array of AAT roles: links between molecular form and function. Diagram created based on data from a previous study [9].

**Figure 3 ijms-21-09187-f003:**
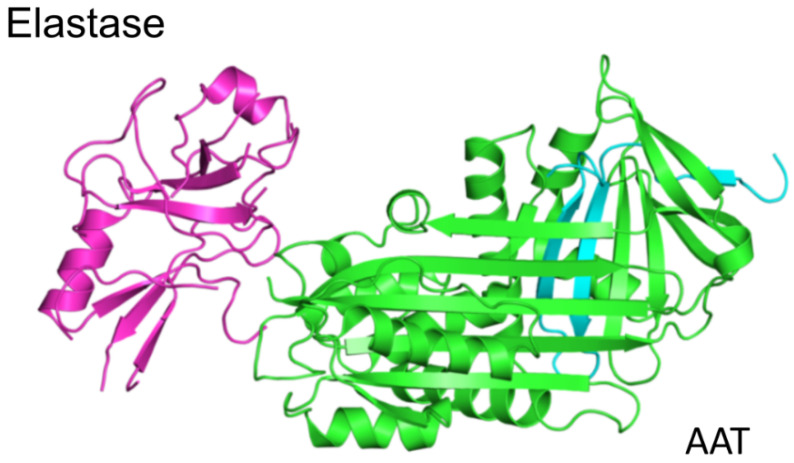
Schematic structure of alpha-1-antitrypsin–protease complex. Conformational transformation is introduced by reaction of the active serine of the protease (elastase) with the reactive center of the serine protease inhibitor (SERPIN) (in AAT). Disruption of the catalytic site limits the ability of the protease to be released from the complex. The AAT–elastase complex in this figure was prepared based on data obtained from the RCSB PDB repository (https://www.rcsb.org last accessed on 23.11.2020).

**Figure 4 ijms-21-09187-f004:**
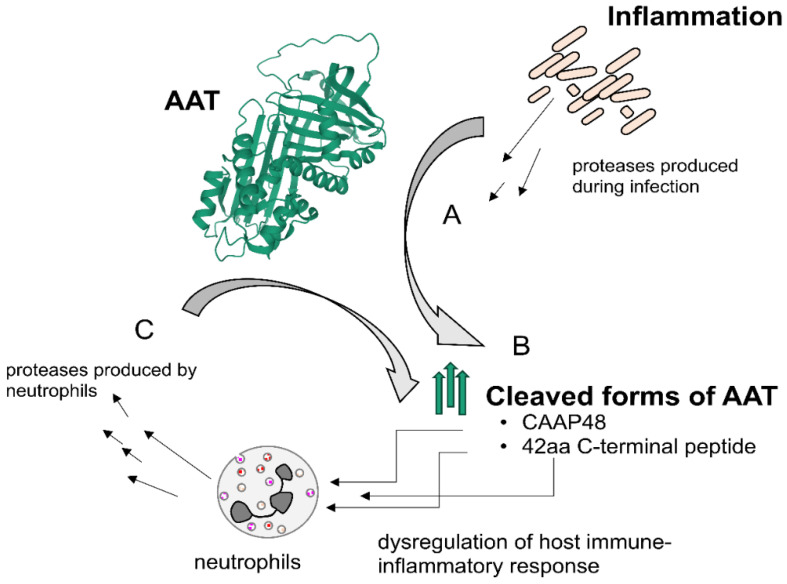
Proposed schematic of the influence of cleaved AAT fragments on the immune–inflammatory response. (**A**). During inflammation (e.g., bacterial infection) protease release leads to AAT cleavage. (**B**). Cleaved forms of AAT activate neutrophils. (**C**). Neutrophils further activate proteases, leading to an enhanced immune response. Diagram modified based on data in Ref. [57]. Abbreviation: 42aa, 42-amino acid.

**Figure 5 ijms-21-09187-f005:**
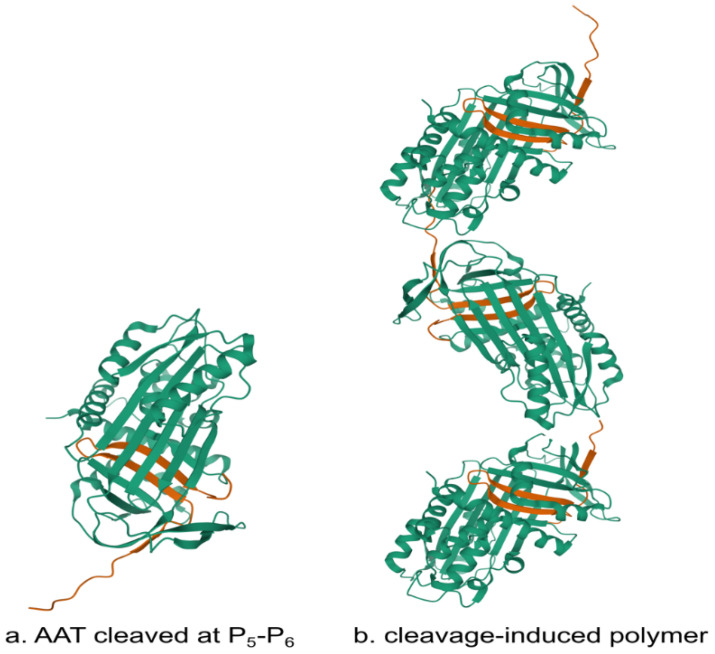
Schematic structures of (**a**) cleaved alpha-1-antitrypsin and (**b**) alpha-1-antitrypsin polymer. These diagrams were prepared based on data obtained from the SWISS-MODEL repository (https://swissmodel.expasy.org last accessed on 23.11.2020).

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
