# Peer review of "Post-Translational Modifications of Circulating Alpha-1-Antitrypsin Protein"

_ijms, 2020, doi:10.3390/ijms21239187_

Round 1

Reviewer 1 Report

Manuscript reference: ijms-1013017

In this review, the authorssummarized the major molecular forms of Alpha1-antitrypsin, including complex, cleaved, glycosylated, oxidized, and S-nitrosylated forms.

I propose some suggestions to improve this manuscript:

It is generally accepted that abbreviations are explained by the first use. There are some abbreviations without the full name, please correct them.

The authors used several old papers(more than 20 years old). I propose to the authors to actualize the bibliography basing on the current scientific evidence.

The authors have to check English throughout the manuscript.

Figures 1, 3and 5are of poorquality;please provide better-quality figures.

Abbreviationlistshould beintroducedbefore introduction,not after the conclusion.

Line 39: “surrounded by circles”, the amino acidsare insquares, not incircles.

Line 239: remove one “that”Line 335: “NO” has to bebetween parentheses.Ipropose to extendsection 6. To give more information about the S-nitrosylation ofAAT under diseases and its effects

Reviewer 2 Report

A review “Post-translational modifications of circulating alpha-2 1-antitrypsin protein” by  Lechowicz is well-written describing the role of Alpha1-antitrypsin protein which is a member of the SERPIN family. The change is the oligomeric status of protein and their association with the various pathological condition is well documented and presented. The effect of PTMs on the regulation of protein functions has been well investigated and shown to associate with the various disease by modulating the functional output of protein.  The information provided in this review is important as well as valuable to general readers. The best part of this review is the information describing the relation of AAT glycosylation and COVID-19 severity. One of the minor concern is poor image quality. I would suggest using a high-resolution color image. The use of different colors for an alpha, Beta, or loop might help to improve the quality and readability of the image. For example, labeling in figure 1 is not readable.
